# A direct RT-qPCR approach to test large numbers of individuals for SARS-CoV-2

**Tomislav Maricic**[1]*, **Olaf Nickel**[2], **Ayinuer Aximu-Petri**[1], **Elena Essel**[1], **Marie Gansauge**[1], **Philipp Kanis**[1], **Dominik Macak**[1], **Julia Richter**[1], **Stephan Riesenberg**[1], **Lukas Bokelmann**[1], **Hugo Zeberg**[1,3], **Matthias Meyer**[1], **Stephan Borte**[2,4,5], **Svante Pääbo**[1]*

**1** Max Planck Institute for Evolutionary Anthropology, Leipzig, Germany, **2** Department of Laboratory Medicine, Hospital St. Georg, Leipzig, Germany, **3** Department of Neuroscience, Karolinska Institutet, Stockholm, Sweden, **4** ImmunoDeficiencyCenter Leipzig (IDCL) at Hospital St. Georg Leipzig, Jeffrey Modell Diagnostic and Research Center for Primary Immunodeficiency Diseases, Leipzig, Germany, **5** Division of Clinical Immunology, Department of Laboratory Medicine, Karolinska University Hospital Huddinge at Karolinska Institutet, Stockholm, Sweden

* tomislav_maricic@eva.mpg.de (TM); paabo@eva.mpg.de (SP)

**Data Availability Statement:** All relevant data are within the manuscript and its Supporting Information files.

## Abstract

SARS-CoV-2 causes substantial morbidity and mortality in elderly and immunocompromised individuals, particularly in retirement homes, where transmission from asymptomatic staff and visitors may introduce the infection. Here we present a cheap and fast screening method based on direct RT-qPCR to detect SARS-CoV-2 in single or pooled gargle lavages ("mouthwashes"). This method detects individuals with large viral loads (Ct≤29) and we use it to test all staff at a nursing home daily over a period of three weeks in order to reduce the risk that the infection penetrates the facility. This or similar approaches can be implemented to protect hospitals, nursing homes and other institutions in this and future viral epidemics.

## Introduction

The COVID-19 pandemic has led to substantial morbidity and mortality in elderly and immunocompromised individuals, particularly in long-term care facilities [1]. Up to half of the fatalities in Europe have been observed in retirement homes [2], where advanced age and chronic underlying health conditions seem to predispose residents [3]. Therefore, it is necessary to prevent that infected personnel and visitors enter institutions where vulnerable individuals live.

However, infected individuals are often asymptomatic [4, 5], resulting in a considerable rate of presymptomatic transmission of SARS-CoV-2. Furthermore, the viral load, and presumably the ability to transmit the infection, is highest at the time of symptom onset or just before [6]. To prevent the spread of infection to institutions, one would therefore ideally test all individuals entering the facilities every time they do so. However, most commercially available viral test systems are too expensive and too laborious for such a strategy. In addition, swab kits and RNA extraction kits may run out during a pandemic [7].

Here, we present a safe, cheap and fast PCR-based approach to detect individuals with high levels of SARS-CoV-2 without the need for nasopharyngeal swabs or RNA extraction. We performed daily tests of all staff and residents of a local nursing home previously affected by

**Funding:** Funding was provided by the Max Planck Society and the NOMIS foundation.

**Competing interests:** The authors have declared that no competing interests exist.

COVID-19. We suggest that this approach could be used to test staff members and visitors of retirement homes and other institutions.

## Results

Nasopharyngeal swabs are commonly used to collect samples for SARS-CoV-2 testing. However, this exposes a person taking the sample to risk of infection when manipulating swabs in the nose and pharynx of potentially infectious individuals. We (in preparation) and others [8–10] have shown that liquid washes of the oral cavity ('gargle lavage') can be used as a more practical, safe yet efficient alternative to nasopharyngeal swabs. This is particularly attractive for frequent sampling of asymptomatic individuals, as gargle lavages are not associated with any discomfort and can be applied to most individuals in a broad age range. We therefore focused our efforts on making such gargle lavages, commonly referred to as 'mouthwashes', amenable to detection of SARS-CoV-2 without isolation of viral RNA.

To this end, we investigated mouthwashes from 10 individuals who previously tested positive for SARS-CoV-2 and 10 individuals who did not. From each mouthwash, we used 200 microliter (μl) for RNA extraction in the MagNA Pure 24 System (Roche, Basel, Switzerland) and 10 μl of the resultant 50 μl RNA extract for reverse transcriptase (RT) quantitative polymerase chain reaction (qPCR) detection of SARS-CoV-2 using the LightCycler Multiplex RNA Master kit from Roche. Nine of the individuals, who on a previous occasion had tested positive, were positive in this occasion whereas none of the previously negative individuals were positive.

We then tested 1 μl of mouthwash from each of the 20 individuals using two RT-qPCR kits advertised to allow direct detection of SARS-CoV-2 from nasopharyngeal swabs: Luna Universal Probe (NEB, Ipswich, USA) and PrimeDirect (Takara, Kyoto, Japan) as well as another kit, SuperScript III with Platinum Taq (Invitrogen, Waltham, USA). The previously negative mouthwashes were negative in all cases. Among the previously positive mouthwashes, five were positive using the NEB kit whereas one and two were positive using the Takara and Invitrogen kits, respectively (Fig 1). Although we expect the tests to be somewhat stochastic at high cycle threshold (Ct) values (the higher the Ct value, the lower the number of viral targets), the NEB Luna kit performs better with mouthwashes than the other two kits.

The Roche RNA extraction and qRT-PCR combination as well as the NEB kit were validated against a set of 7 reference samples distributed by "INSTAND" reference institution of the German Medical Association (Düsseldorf, Germany). These samples contain different concentrations of a heat-inactivated SARS-CoV-2 virus (sample 59, 61, 63 and 64), another

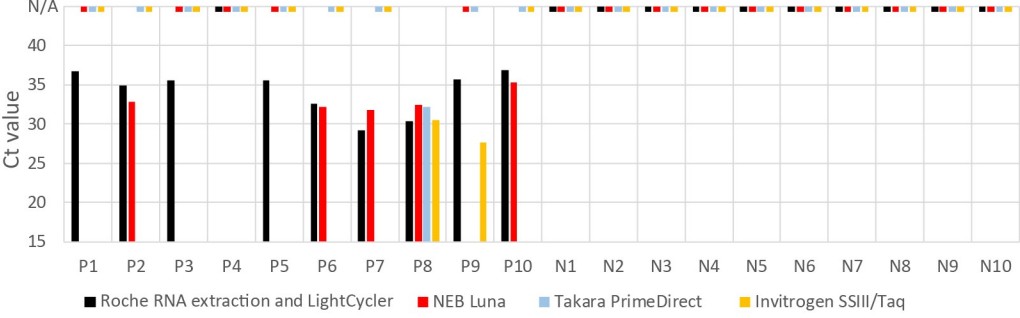

**Fig 1. RT-qPCR of 10 previously SARS-CoV-2 positive (P) and 10 previously SARS-CoV-2 negative (N) mouthwashes.** RNA was either extracted from 200 μl of mouthwashes (black bars) or 1 μl of the mouthwashes was added directly to the RT-qPCR reaction mix without prior extraction (colored bars). Dashes at N/A indicate no amplification of the viral target.

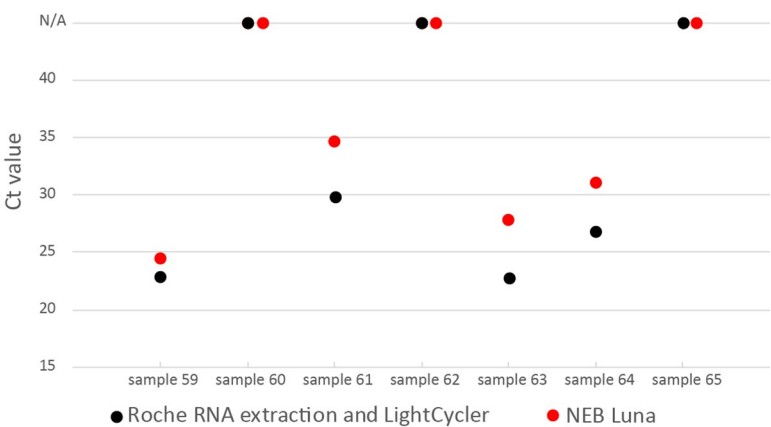

**Fig 2. Roche RNA extraction/RT-qPCR and NEB Luna RT-qPCR of seven reference samples.** Dots at N/A show no amplification of the viral target.

coronavirus genus (sample 60 HCoV OC43, sample 65 HCoV 229E) or no virus (sample 62, supernatant of a non-infected cell culture). Fig 2 shows that both approaches detected the SARS-CoV-2-containing samples and showed not amplification for samples not containing SARS-CoV-2.

To systematically investigate how the NEB Luna assay performs compared to RNA extraction followed by the Roche assay for mouthwash samples, we investigated 62 gargle lavages from patients that were either negative or presented with various viral loads based on previous investigations. Fig 3A shows that although the direct assay requires about four additional qPCR cycles on average to detect the virus, the direct assay performs well in samples where the RNA extraction-based assay detects viruses at a Ct values of ~29 or lower, *i.e.* for samples that contain high viral genome counts. Among the samples positive in both methods, *i.e.* samples with Ct values of ~40 or lower for the NEB assay, the results of the two methods correlate reasonably well (Fig 3B; Pearson's r = 0.86). Notably, the extraction-free NEB assay uses 20-fold less mouthwash per assay than the Roche assay (2 µl vs. 40 µl), corresponding approximately to the average difference of the observed four PCR cycles ($2^4 = 16$). Thus, the NEB assay *per se* is of similar sensitivity as RNA extraction followed by the Roche assay. Next we tested the smallest number of SARS-CoV-2 RNA molecules that can be detected in the assay. We find that six RNA molecules can be detected in 19 out of 20 replicate reactions, which corresponds to limit of detection of 3,000 RNA molecules per milliliter of mouthwash (S1 Table).

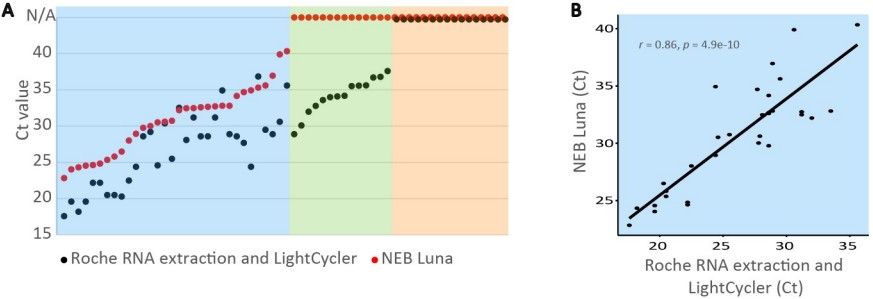

**Fig 3.** (**A**) Roche RNA extraction/RT-qPCR and NEB Luna RT-qPCR for SARS-CoV-2 detection in 62 mouthwash samples from hospital patients. The NEB method detects the virus when Ct of the Roche method is less than 29 (blue box). Dots at N/A show no amplification of the viral target. (**B**) Correlation of Ct values between direct NEB assays and RNA extraction followed by Roche assays for mouthwash samples positive with both methods.

We applied this testing approach to the residents and personnel of the retirement home Emmaus in Leipzig, Germany. This facility has 77 full-time residents and 88 nursing, cleaning and catering staff. In April 2020, two COVID-19 cases occurred among the residents and staff.

After appropriate informed consent, all staff working at the facility delivered a daily gargle lavage sample at the beginning or end of their working shifts. These were collected between 3 p.m. and 4 p.m. every day for 21 days and transported to the laboratory at room temperature. On weekends and holidays, the numbers of samples varied between 22 and 26 and during weekdays between 35 and 50. We applied two testing schemes to minimize the risk that the virus could affect residents again. In the first scheme, the samples were tested individually using the direct RT-qPCR protocol and the results were evaluated and reported back to the facility by 7 p.m. To detect any inhibition that the mouthwash samples may introduce into the RT-qPCR reactions, we added a synthesized control RNA that was quantified in parallel with SARS-CoV-2 by a probe carrying a different fluorophore (Cy5) from the SARS-CoV-2 probe (FAM). For three of the 756 mouthwashes, significant inhibition was detected (S2 Table). These samples were further tested using half the mouthwash volume, which lowered the inhibition to acceptable levels (S2 Table). Nine samples were putatively positive at Ct values of 36 and higher. These were further tested by agarose electrophoresis of the PCR product, adding an hour to the analysis time, and by further RT-qPCR tests. In all cases, the PCR products were of unexpected sizes and renewed testing was negative (S2 Table). In summary, all 756 individual mouthwashes we tested over 21 days were negative.

In a second scheme (Fig 4), we explored if the daily testing could be scaled up by pooling samples. To this end, we first investigated if single positive samples with Ct values between 19 and 31 could be detected in pools of 26 samples. From these pools, 2 µl were used for the direct NEB assay while 200 µl were used for the RNA extraction and the Roche assay. Fig 5 shows that in all six cases, the pools containing a mouthwash from a single infected individual were detected by the extraction-free NEB assay as well as by the standard Roche assays after RNA extraction.

We combined the samples from the Emmaus retirement home from each day into two pools of approximately equally many individuals and tested these pools (Fig 4). In agreement with the individual test, none of the pools were positive. Ct values for the positive RNA control was ~27 when no mouthwash was added and ~28 or less for the pools, indicating that the inhibitory component seen in rare individual mouthwash samples becomes diluted and do not inhibit the reaction as much in a pool.

## Discussion

During the current SARS-CoV-2 epidemic morbidity and mortality primarily affect old or immunocompromised individuals. Thus, it is important to protect facilities where such individuals live from infections transmitted by staff or visitors. This applies not only to SARS-CoV-2 but also to other airborne communicable diseases such as influenza, RSV and adenovirus, which result in high rates of disease and death among vulnerable individuals.

To protect such individuals, it is necessary to reduce the risk of transmission from asymptomatic carriers. This is particularly the case for SARS-CoV-2, as many carriers are asymptomatic [4], and their infectiousness may be highest before the onset of symptoms [6, 11]. It is therefore desirable to test asymptomatic individuals that come into contact with vulnerable individuals as frequently as possible [12, 13]. The protocol described here enables daily testing of substantial numbers of individuals for screening purposes at a reagent costs of about 2–4 Euros per sample (NEB Luna kit ~0.6 Euros plus primer/probe 1.5–3.5 Euros per sample).

nursing home

77 staff    88 residents

21 days

22-50 employees
per day

mouthwash collection

transport to laboratory

optional pooling of
up to 25 mouthwashes
per pool

RT-qPCR

evaluation

reporting

**Fig 4. Daily testing of personnel of a nursing home with or without pooling of mouthwashes.** Red color indicate a SARS-CoV-2 positive individual/sample.

Three factors make this approach suitable for screening. Firstly, gargle washes rather than nasopharyngeal swabs can be collected by the individuals themselves and do not expose personnel performing sampling to risk of infection. They are also not associated with any discomfort for the persons being tested so that testing can be performed on a daily basis. Secondly, since the protocol does not require isolation of RNA, it reduces the need for equipment, reagents, labor, time and money. Thirdly, pooling of samples allows large numbers of individuals to be tested. A practical approach might be to divide the staff at each institution into two or more pools to allow some groups to return to work even if one group turns out to contain one or more positive individuals.

Nine out of 756 mouthwashes analyzed were positive at Ct values above 36. Agarose gel electrophoresis, which can be done within an hour, as well as retesting, showed that these were false positives. This issue can be eliminated by the addition of an additional primer pair and probe in the qPCR reaction (nCOV_N1, HEX-fluorophore, approved for SARS-CoV-2 testing by Centers for Disease Control and Prevention, USA) and requiring both probes to emit fluorescence to declare a sample positive (data not shown).

## Limitations and practical considerations

The direct qPCR assay described is slightly less sensitive than assays that require RNA isolation due to the smaller volume of mouthwash that can be used in the reaction. However, there is evidence that only individuals with a high viral load (a Ct value of 30 or lower) are likely to transmit the infection [11, 14, 15]. Thus, as has also been suggested by others [12, 13], we

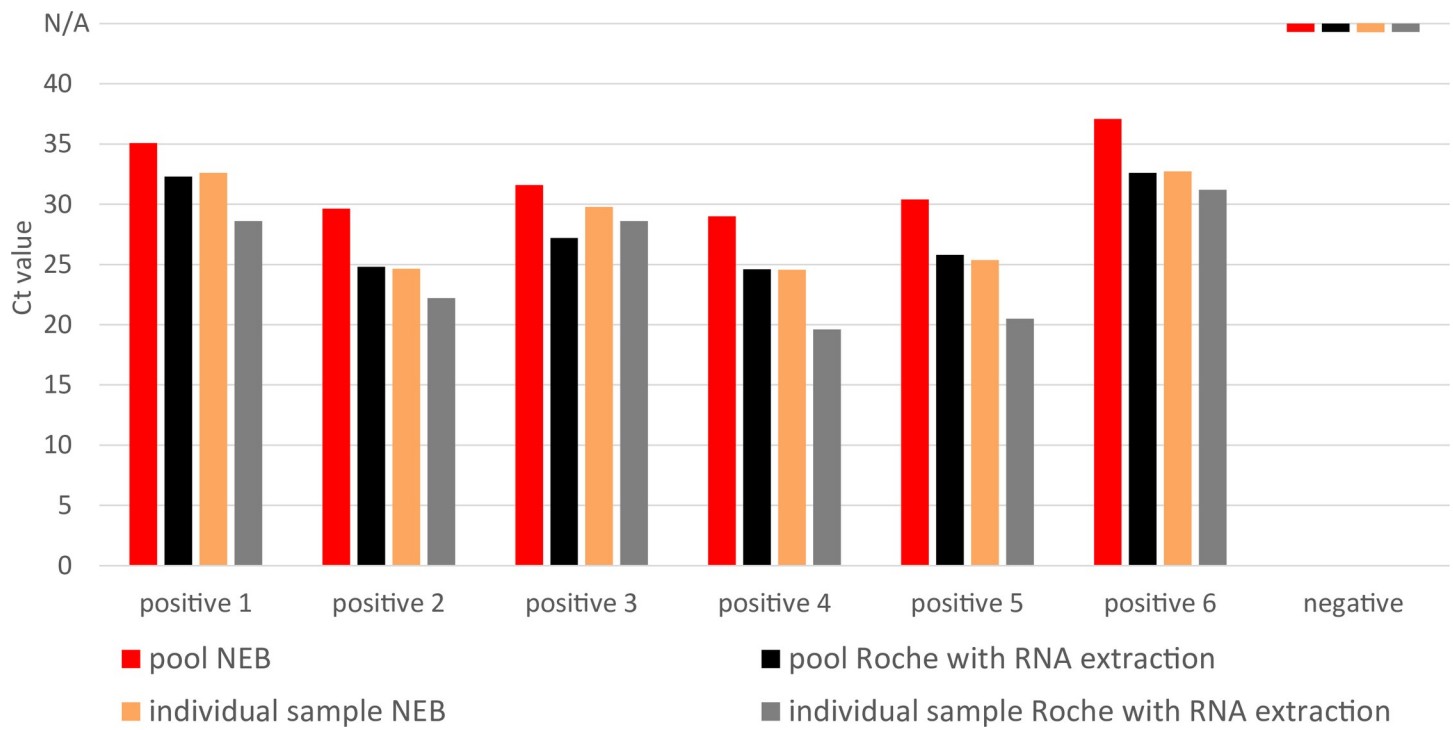

**Fig 5. Tests of seven pools of 26 mouthwash samples, which in six cases contain one positive sample.** Dashes at N/A show no amplification of the viral target.

suggest that the advantage of daily testing and fast turn-around times outweighs the disadvantage of the lower sensitivity. Currently, we perform tri-weekly test of an average of 150 of employees per day at our research institute. Two people do the pooling of samples and set-up of the RT-qPCR in 40 minutes. Evaluations of results and reporting follows two hours later. When individual tests need to be performed, they require another 2–3 hours.

We note that rapid and inexpensive testing schemes as the one presented here may contribute to making visits to individuals at risk possible if visitors are tested within a few hours before visits. They could also be applied to travelers and personnel necessary for critical infrastructure.

## Materials and methods

All individuals included in this study were asked for their voluntary assistance to participate. Each individual gave written informed consent before entry into the study. The study was approved by the Ethics Committee of the Saxonian medical chamber (EK-allg-37/10-1). All procedures utilized in this study are in agreement with the 1975 Declaration of Helsinki.

### Gargle lavages ('mouthwashes')

For mouthwashes, 10 mL of sterile water was pre-filled into sterile scaled urine cups. The individuals to be tested took the full volume of water into the mouth and gargled for 10 seconds. Subsequently, the mouthwash was spitted back into the cup and firmly closed with a watertight lid and stored at room temperature for up to 4–8 hours. We also analyzed anonymized leftover material from hospitalized COVID-19 patients. In these cases, mouthwashes had been stored frozen below -80°C for 1–2 months.

### Primers and probes for RT-qPCR

All primers, probes and controls were from Tib-Molbiol (Berlin, Germany). For SARS-CoV-2 detection, the E-gene was detected with a FAM probe and primers from a premixed kit (Cat.-No. 53-0776-96) as described in [16] (E_Sarbeco_F `ACAGGTACGTTAATAGTTAATAGCGT`, E_Sarbeco_P1 `FAM-ACACTAGCCATCCTTACTGCGCTTCG-BBQ`, E_Sarbeco_R `ATATTG CAGCAGTACGCACACA`). For inhibition tests, EAV RNA Extraction Control 660 with Cy5 probe and primers from the kit (Cat.-No. 66-0909-96) were used.

### Roche RNA extraction and RT-qPCR

Individual mouthwashes were vortexed and 200 µl were used for RNA extraction with the MagNA Pure 24 System from Roche based on the manufacturer's protocol. RNA was eluted in 50 µl of which 10 µl was used in the RT-qPCR LightCycler Multiplex RNA Master kit with 4.9 µl water, 0.5 µl E-gene primer and probe reagent mix, 0.5 µl EAV control primer and probe reagent mix, 4 µl RT-PCR reaction mix and 0.1 µl RT-enzyme solution. The RT reaction was performed at 55°C for 5 minutes. qPCR included a denaturation step at 95°C for 5 minutes, followed by 45 cycles with 5 seconds at 95°C, 15 second at 60°C and 15 seconds at 72°C. RT-qPCR reaction was performed on a Cobas z480 Analyzer and analyzed with Roche LightCycler 480 Software.

### Direct RT-qPCR

Mouthwashes were vortexed and either 1 or 2 µl were added into a reaction tube containing pre-assembled reagents. RT-qPCR reaction was run on BioRad CFX96 Real-Time system and

results analyzed with Bio-Rad CFX Manager software where the Ct determination mode was set to "regression".

### Takara PrimeDirect Probe one-step RT-qPCR

Twenty five µl reactions contained: 12.5 µl prime direct mix, 0.5 µl E-gene primer and probe reagent mix, 1 µl mouthwash and 11 µl water. The reaction was incubated at 90˚C for 3 minutes, 60˚C for 5 minutes followed by 50 cycles of 95˚C for 5 seconds and 55˚C for 30 seconds.

### Invitrogen SuperScript III with Platinum Taq one-step RT-qPCR

Twenty five µl reaction contained: 12.5 µl rxn mix, 0.5 µl E-gene primer and probe reagent mix, 0.4 µl MgSO4, 1 µl enzyme mix, 1 µl mouthwash and 9.6 µl water. The reaction was incubated at 60˚C for 5 minutes, 55˚C for 5 minutes, 95˚C for 2 minutes followed by 50 cycles of 94˚C for 5 seconds, 60˚C for 30 seconds and 68˚C for 10 seconds.

### NEB Luna Universal Probe one-step RE-qPCR kit

Twenty five µl reaction contained: 12.5 µl Reaction MasterMix, 1.25 µl EnzymeMix, 0.5 µl E-gene primer and probe reagent mix, 0.5 µl EAV control primer and probe reagent mix, 0.5 µl EAV control (pellet eluted in 1,200 µl TE buffer), 1–2 µl mouthwash, and 8.75–9.75 µl water. For Fig 1, 1 µl of mouthwash was used, for later experiments 2 µl to allow for more accurate pipetting. Mastermix can be stored in the fridge up to a week. RT-qPCR program for the NEB reaction was: 55˚C for 4 minutes, 60˚C for 4 minutes, 55˚C for 2 minutes, 95˚C for 1 minute followed by 50 cycles of 95˚C for 10 seconds and 60˚C for 30 seconds.

## Supporting information

**S1 Table. Analytical limit of detection of the gurgle NEB SARS-CoV-2 assay.**
(DOCX)

**S2 Table. Ct values and gel pictures.**
(XLSX)

**S1 Raw images.**
(PDF)

## Acknowledgments

The authors thank the management of Diakonische Dienste Leipzig, the personnel and residents of Altenpflegeheim Emmaus in Leipzig, Germany, for their cooperation. Furthermore, we recognize the efforts and helpful instructions contributed by Nils Lahl from the public health department of the City of Leipzig. We particularly thank Jeannine Dulz, Katrin Grabietz, Jeannette Hofmann, Michelle Karius, Sandra Reinhardt, Kerstin Rolle, Katharina Wald, Kristina Winter for assisting and performing reference and confirmatory testing at the Department of Laboratory Medicine at Hospital St Georg Leipzig.

## Author Contributions

**Conceptualization:** Tomislav Maricic, Matthias Meyer, Svante Pääbo.

**Data curation:** Tomislav Maricic, Ayinuer Aximu-Petri.

**Formal analysis:** Tomislav Maricic, Olaf Nickel.

**Funding acquisition:** Svante Pääbo.

**Investigation:** Tomislav Maricic, Olaf Nickel, Ayinuer Aximu-Petri, Stephan Riesenberg, Lukas Bokelmann.

**Methodology:** Tomislav Maricic, Stephan Riesenberg, Lukas Bokelmann.

**Project administration:** Tomislav Maricic, Stephan Borte.

**Resources:** Olaf Nickel, Stephan Borte.

**Supervision:** Tomislav Maricic, Matthias Meyer, Svante Pääbo.

**Validation:** Tomislav Maricic, Olaf Nickel, Ayinuer Aximu-Petri, Elena Essel, Marie Gansauge, Philipp Kanis, Dominik Macak, Julia Richter.

**Visualization:** Tomislav Maricic.

**Writing – original draft:** Tomislav Maricic, Hugo Zeberg, Stephan Borte, Svante Pääbo.

**Writing – review & editing:** Tomislav Maricic, Stephan Borte, Svante Pääbo.

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
