## [Decision Letter · Decision Letter 0]

2 Nov 2020

PONE-D-20-23519

A direct RT-qPCR approach to test large numbers of individuals for SARS-CoV-2

PLOS ONE

Dear Dr. Maricic,

Thank you for submitting your manuscript to PLOS ONE. After careful consideration, we feel that it has merit but does not fully meet PLOS ONE’s publication criteria as it currently stands. Therefore, we invite you to submit a revised version of the manuscript that addresses the points raised during the review process.

We look forward to receiving your revised manuscript.

Kind regards,

Ronald Dijkman, PhD

Academic Editor

PLOS ONE

Additional Editor Comments:

Dear Dr. Maricic,

My apologies for the long duration of the reviewing process, however, it has been very difficult to find the appropriate number of reviewers to evaluate your work. Based on the reviewer comments (No. 2 & 3), is that the study lacks extensive validation of the direct RT-PCR protocol. After reading your manuscript, I tend agree with the reviewers that in the current manuscript version is not well addressed, and therefore I strongly encourage you to address this critical point.

Best regards,

Ronald Dijkman

Journal Requirements:

"Funding was provided by the Max Planck Society and the NOMIS foundation."

Reviewers' comments:

Reviewer's Responses to Questions

**Comments to the Author**

1. Is the manuscript technically sound, and do the data support the conclusions?

Reviewer #1: Yes

Reviewer #2: No

Reviewer #3: Yes

2. Has the statistical analysis been performed appropriately and rigorously? 

Reviewer #1: I Don't Know

Reviewer #2: N/A

Reviewer #3: Yes

3. Have the authors made all data underlying the findings in their manuscript fully available?

Reviewer #1: Yes

Reviewer #2: Yes

Reviewer #3: Yes

4. Is the manuscript presented in an intelligible fashion and written in standard English?

Reviewer #1: Yes

Reviewer #2: Yes

Reviewer #3: Yes

5. Review Comments to the Author

Reviewer #1: The authors present a performance evaluation of different RT PCR-based detection methods for Sars-CoV-2. The main advantages of two new protocols is the use of gargles instead of nasopharyngeal swab material and poolin. They show acceptable detection performance.

Major comments:

none.

Minor comments: Ct value, Cq value are not consistently used. élease use same wording throughout.

Reviewer #2: Maricic et al. present a direct RT-qPCR approach to test large number of individuals for SARS-CoV-2. Direct RT-qPCR testing strategies are important to mitigate supply chain issues and lower cost and time for SARS-CoV-2 testing. Although the authors present an important proof of concept, several similar direct protocols have already been published and the current study lacks extensive validation of the protocol to support its major claims.

Major

1. Several studies have shown that gargle lavages can be used to detect SARS-CoV-2, as well as extraction-free/direct RT-qPCR assays and therefore the originality and novelty of this study is limited.

2. The direct RT-qPCR assay has not been extensively validated. Overall, sample size for clinical testing is low, and specificity (and cross-reactivity) is only tested with 7 reference samples. The authors did test a substantial number of samples collected in the nursing home, but out of the 756 samples tested, none of those were positive. Based on these results test performance cannot be evaluated in the target population, and therefore the main claim that this method can be used for testing in nursing homes cannot be supported.

3. Sensitivity for the direct RT-qPCR assay was not determined and based on the presented data it seems like this assay can only be used for samples with a Ct value below 29 which is not sensitive enough to detect cases beyond peak viral titers. The limit of detection is an important parameter to determine the performance of a test.

Minor

4. Please include primer and probe sequences in the methods section.

Reviewer #3: Well performed study looking at a novel collection sample (gargle washes) with direct RT-PCR, as well as pilot study of pooling these samples in this test system. The methodology is sound and the conclusions warranted by the data, but a lack of context limits the value of the conclusions, and the discussion is overspeculative.

-Describing this methodology as large-scale high-throughput is a considerable stretch. The authors' actual testing performance was 35-50 samples per weekday, 22-26 on weekends; with no positive samples. While scaling-up by pooling in theory could provide '~3,700 individuals, with ~300 Euros total reagent cost in an hour and a half', this scaling-up was not field tested. There is considerable distance between a research-ready and a clinically-ready process.

-Labor is as limiting for many testing laboratories as are reagents. Please describe the setup time per run for the unpooled and pooled versions of the assay.

-The operational discussion of assay use in long-term care (lines 189-197) hopelessly oversimplifies the logistics of such an approach. Recommend omitting.

-The discussion in lines 203-211 is plausible but, at this time, rather contentious and by no means established science. Interruption of COVID transmission by large-scale rapid testing is conceivable, but faces formidable logistical and behavioral barriers. Most discussions of such approaches focus on home-performed testing. The authors of this study, again, fail to recognize the challenges of scaling-up centralized testing, even with simplified lab-based protocols such as they describe. A proper discussion of this topic is beyond the scope of the paper; the authors should at least soften this paragraph considerably.

-Lines 212-215 speculate excessively. Please omit.

-The specificity of the test appears to be roughly 98.8% (9 false-positives out of 756 samples tested). How does this impact usability and scaleability? How complex and time-consuming are the described procedures for assessing false-positives? Lines 136-138

-There is something of an imbalance between the description of this testing methodology as simple, rapid, and scaleable; which requires the gavage sample type; and the paucity of data on the clinical performance of the gavage. only ten positive samples were assessed; and the clinical sensitivity appears to be around 50%. The samples missed were high Ct value samples of (probably) modest infection-control significance; but 50% is 50% and only-ten samples are only-ten samples. I recognize that Germany during the period of this study was a low-incidence area; but the conclusions must be tempered by this major weakness.

6. PLOS authors have the option to publish the peer review history of their article (what does this mean?). If published, this will include your full peer review and any attached files.

Reviewer #1: **Yes: **Peter M. Keller

Reviewer #2: No

Reviewer #3: **Yes: **Sheldon Campbell

---

## [Author Response · Author response to Decision Letter 0]

25 Nov 2020

Additional Editor Comments:

Dear Dr. Maricic,

My apologies for the long duration of the reviewing process, however, it has been very difficult to find the appropriate number of reviewers to evaluate your work. Based on the reviewer comments (No. 2 & 3), is that the study lacks extensive validation of the direct RT-PCR protocol. After reading your manuscript, I tend agree with the reviewers that in the current manuscript version is not well addressed, and therefore I strongly encourage you to address this critical point.

We believe that the issue about validation is at least partially a misunderstanding. We have tested much more than the ten positive and ten negative samples to which reviewer 3 refers. We tested 756 mouthwash samples in the nursing home showing that the false positive rate is very low (Results section, line 139-140). We also tested 46 samples previously been shown to be positive and show that we detect all samples with Ct values at and below 29. See answers to the reviewers below.

We may also mention that we now perform between 400 and 500 tests per week in our institute. This method is also being established at other institutions in Germany. 

We strongly believe that it will be useful for the manuscript to be published at this point in time.

"Funding was provided by the Max Planck Society and the NOMIS foundation."

We have deleted this sentence in the Acknowledgments section.

The sentence is OK.

A supplementary file with uncropped and unadjusted images has now been uploaded.

This has now been done.

Reviewers' comments:

Reviewer's Responses to Questions

Comments to the Author

1. Is the manuscript technically sound, and do the data support the conclusions?

Reviewer #1: Yes

Reviewer #2: No

Reviewer #3: Yes

2. Has the statistical analysis been performed appropriately and rigorously?

Reviewer #1: I Don't Know

Reviewer #2: N/A

Reviewer #3: Yes

3. Have the authors made all data underlying the findings in their manuscript fully available?

Reviewer #1: Yes

Reviewer #2: Yes

Reviewer #3: Yes

4. Is the manuscript presented in an intelligible fashion and written in standard English?

Reviewer #1: Yes

Reviewer #2: Yes

Reviewer #3: Yes

5. Review Comments to the Author

Reviewer #1: The authors present a performance evaluation of different RT PCR-based detection methods for Sars-CoV-2. The main advantages of two new protocols is the use of gargles instead of nasopharyngeal swab material and pooling. They show acceptable detection performance.

Major comments:

none.

Minor comments: Ct value, Cq value are not consistently used. élease use same wording throughout.

We changed all Cq to Ct in the text and the figures.

Reviewer #2: Maricic et al. present a direct RT-qPCR approach to test large number of individuals for SARS-CoV-2. Direct RT-qPCR testing strategies are important to mitigate supply chain issues and lower cost and time for SARS-CoV-2 testing. Although the authors present an important proof of concept, several similar direct protocols have already been published and the current study lacks extensive validation of the protocol to support its major claims.

Similar direct RT-qPCR tests have been published but our protocol is a unique combination of using mouthwashes, direct RT-qPCR and pooling. This approach allows substantial numbers of individuals to be screened and individuals with high viral loads to be detected in a short time.

We now stress in the paper that this protocol is a screening or “surveillance” approach. It is in the meantime implemented in in our institute as well as other institutions. 

Major

1. Several studies have shown that gargle lavages can be used to detect SARS-CoV-2, as well as extraction-free/direct RT-qPCR assays and therefore the originality and novelty of this study is limited.

Our protocol is a unique combination of using gargle lavages, direct RT-qPCR, and pooling. It is a surveillance method that in our hands is currently proving valuable in the second wave of infections. We believe that it may be useful also to others.

2. The direct RT-qPCR assay has not been extensively validated. Overall, sample size for clinical testing is low, and specificity (and cross-reactivity) is only tested with 7 reference samples. The authors did test a substantial number of samples collected in the nursing home, but out of the 756 samples tested, none of those were positive. Based on these results test performance cannot be evaluated in the target population, and therefore the main claim that this method can be used for testing in nursing homes cannot be supported.

The test with the 7 reference samples of which two contain another coronavirus genus (Figure 2) is a test that SARS-CoV-2 diagnostic lab in Germany has to pass. Cross-reactivity of the primers/probe combination was extensively tested in the first publication of the E-gene primers and is recommended by the WHO. We now cite this first publication the E-gene primers by Corman et al. in the Methods section. 

3. Sensitivity for the direct RT-qPCR assay was not determined and based on the presented data it seems like this assay can only be used for samples with a Ct value below 29 which is not sensitive enough to detect cases beyond peak viral titers. The limit of detection is an important parameter to determine the performance of a test.

As the reviewer points out, Figure 3 shows that this protocol reliably detects samples with a Ct value of 29 and lower. As we point out in Discussion section (lines 191-192), it is thus slightly less sensitive than the standard assay. However, as we now stress in the manuscript, this is a surveillance method, designed to detect individuals likely to transmit the virus. As we also point out in the manuscript (lines 192-194), there is evidence that only individuals with Ct of 30 or lower are likely to transmit the infection (NCID, 2020; Wolfel et al, 2020; Zhou et al, 2020). For example, the position statement from National Centre for Infectious Diseases Singapore (NCID, 2020) states: "when the Ct value was 30 or higher (i.e. when viral load is low), no viable virus (based on being able to culture the virus) has been found”. 

This is also stressed in a recent perspectives piece in NEJM (https://www.nejm.org/doi/full/10.1056/NEJMp2025631) which states: "effective surveillance regimens intended to reduce the population prevalence of a respiratory virus need to return results quickly to limit asymptomatic spread and should be sufficiently inexpensive and easy to execute to allow frequent testing”. They also say there "we need tests that can enable regimens that will capture *most* infections while they are still infectious." 

Another recent study model surveillance effectiveness considering test sensitivities, frequency, and sample-to-answer reporting time (https://www.ncbi.nlm.nih.gov/pmc/articles/PMC7325181.2/) and conclude that: “surveillance should prioritize accessibility, frequency, and sample-to-answer time; analytical limits of detection should be secondary”.

We believe that our approach fulfills these needs for surveillance (daily testing, results in 5 hours, detection individuals with high viral loads (Ct <29). 

To make this clear, we now stress that our protocol is a surveillance strategy rather than a diagnostic procedure throughout the manuscript. 

Minor

4. Please include primer and probe sequences in the methods section.

The probes and primers for the E-gene is now the Methods section (line 221-225). For the internal RT-qPCR control (EAV) we give the catalog number since the producer does not reveal its sequence (line 225-226. 

Reviewer #3: Well performed study looking at a novel collection sample (gargle washes) with direct RT-PCR, as well as pilot study of pooling these samples in this test system. The methodology is sound and the conclusions warranted by the data, but a lack of context limits the value of the conclusions, and the discussion is overspeculative.

We have tried to include more context and reduce speculation in the discussion. In particular, we now stress that that what we present is a protocol that allows surveillance of personnel and residents in institutions with rapid turn-around times and detection in individuals with viral loads high enough to make them likely to be infectious.

-Describing this methodology as large-scale high-throughput is a considerable stretch. The authors' actual testing performance was 35-50 samples per weekday, 22-26 on weekends; with no positive samples. While scaling-up by pooling in theory could provide '~3,700 individuals, with ~300 Euros total reagent cost in an hour and a half', this scaling-up was not field tested. There is considerable distance between a research-ready and a clinically-ready process.

We agree that the protocol is not extensively field-tested. We have tempered our perhaps over- enthusiastic language throughout the manuscript. We note though, that this protocol is now used for tri-weekly testing of the ~300 employees at our institute. It is currently being implemented at other institutes in Germany.

-Labor is as limiting for many testing laboratories as are reagents. Please describe the setup time per run for the unpooled and pooled versions of the assay.

We now included this paragraph in the Discussion discussing this (line 196-199):

 “Currently, we perform tri-weekly test of an average of 150 of employees per day at our research institute. Two people do the pooling of samples and set-up of the RT-qPCR in 40 minutes. Evaluations of results and reporting follows two hours later. When individual tests need to be performed, they require another 2-3 hours.”

-The operational discussion of assay use in long-term care (lines 189-197) hopelessly oversimplifies the logistics of such an approach. Recommend omitting.

We have omitted this paragraph as suggested. 

-The discussion in lines 203-211 is plausible but, at this time, rather contentious and by no means established science. Interruption of COVID transmission by large-scale rapid testing is conceivable, but faces formidable logistical and behavioral barriers. Most discussions of such approaches focus on home-performed testing. The authors of this study, again, fail to recognize the challenges of scaling-up centralized testing, even with simplified lab-based protocols such as they describe. A proper discussion of this topic is beyond the scope of the paper; the authors should at least soften this paragraph considerably.

We have now softened the paragraph by saying that we speculate that this may be possible. 

-Lines 212-215 speculate excessively. Please omit.

We have omitted the paragraph as suggested, although we do not find it excessive to speculate that more lethal viral epidemics may appear in the future. 

-The specificity of the test appears to be roughly 98.8% (9 false-positives out of 756 samples tested). How does this impact usability and scaleability? How complex and time-consuming are the described procedures for assessing false-positives? Lines 136-138

We now clarify that the additional electrophoretic assay requires less than one hour. However, in the meantime, we have eliminated this issue by adding an additional primer pair and a probe (nCOV_N1, approved for SARS-CoV-2 testing by Centers for Disease Control and Prevention, USA) that target another part of the SARS-CoV-2 genome and requiring both probes to emit fluorescence. This is now mentioned in the Discussion section (line 185-190).

-There is something of an imbalance between the description of this testing methodology as simple, rapid, and scaleable; which requires the gavage sample type; and the paucity of data on the clinical performance of the gavage. only ten positive samples were assessed; and the clinical sensitivity appears to be around 50%. The samples missed were high Ct value samples of (probably) modest infection-control significance; but 50% is 50% and only-ten samples are only-ten samples. I recognize that Germany during the period of this study was a low-incidence area; but the conclusions must be tempered by this major weakness.

We tested more than the ten positive and ten negative samples to which the reviewer refers. We tested 46 samples previously been shown to be positive (shown in Fig. 3). We detect all samples with Ct <29. We also tested 756 mouthwash samples in the nursing home which were negative (Results, line 139-140). 

In the meantime, this protocol is being used for tri-weekly testing of the ~300 employees at our institute and the test is currently being implemented at other institutes in our region.

We have tempered the conclusions somewhat as suggested, but remain convinced that what is described here is making a useful contribution towards limiting the spread of infections.

6. PLOS authors have the option to publish the peer review history of their article (what does this mean?). If published, this will include your full peer review and any attached files.

Do you want your identity to be public for this peer review? For information about this choice, including consent withdrawal, please see our Privacy Policy.

Reviewer #1: Yes: Peter M. Keller

Reviewer #2: No

Reviewer #3: Yes: Sheldon Campbell

---

## [Decision Letter · Decision Letter 1]

9 Dec 2020

PONE-D-20-23519R1

A direct RT-qPCR approach to test large numbers of individuals for SARS-CoV-2

PLOS ONE

Dear Dr. Maricic,

Thank you for submitting your manuscript to PLOS ONE. After careful consideration, we feel that it has merit but does not fully meet PLOS ONE’s publication criteria as it currently stands. Therefore, we invite you to submit a revised version of the manuscript that addresses the points raised during the review process.

We look forward to receiving your revised manuscript.

Kind regards,

Ronald Dijkman, PhD

Academic Editor

PLOS ONE

Reviewers' comments:

Reviewer's Responses to Questions

**Comments to the Author**

1. If the authors have adequately addressed your comments raised in a previous round of review and you feel that this manuscript is now acceptable for publication, you may indicate that here to bypass the “Comments to the Author” section, enter your conflict of interest statement in the “Confidential to Editor” section, and submit your "Accept" recommendation.

Reviewer #1: All comments have been addressed

Reviewer #2: (No Response)

Reviewer #3: All comments have been addressed

2. Is the manuscript technically sound, and do the data support the conclusions?

Reviewer #1: Yes

Reviewer #2: Partly

Reviewer #3: Yes

3. Has the statistical analysis been performed appropriately and rigorously? 

Reviewer #1: N/A

Reviewer #2: N/A

Reviewer #3: Yes

4. Have the authors made all data underlying the findings in their manuscript fully available?

Reviewer #1: Yes

Reviewer #2: Yes

Reviewer #3: Yes

5. Is the manuscript presented in an intelligible fashion and written in standard English?

Reviewer #1: Yes

Reviewer #2: Yes

Reviewer #3: Yes

6. Review Comments to the Author

Reviewer #1: The authors have addressed my comments of review round #1. The comments regarding the validation of the RT-PCR protocol raised by reviewer #2 have been addressed in the response to reviewer letter by the authors.

Reviewer #2: I would like to thank the authors for revising their manuscript. The authors have partly addressed my comments, and I still have 2 remaining comments:

1. In their responses to the other reviewers and my comments, the authors state that their proposed protocol is for SARS-CoV-2 surveillance. However, I think the authors have mixed up the meaning of surveillance and screening, which refer to different types of tests in epidemiology (https://www.fda.gov/medical-devices/coronavirus-covid-19-and-medical-devices/covid-19-test-uses-faqs-testing-sars-cov-2). Surveillance is applied to the population level and not to report results back to the individual level, while screening refers to testing of individuals without symptoms. Thus, the authors should use the proper terminology throughout their manuscript.

2. In my previous review I commented on the lack of an extensive sensitivity analysis for the current protocol. I agree with the authors that screening assays may need a different level of sensitivity as compared to diagnostic assays. However, sensitivity analysis is important to enable comparisons between assays to help labs make a decision for an assay. For rapid tests and at-home test I think lower sensitivity is justified, but considering that the current assay is PCR-based and that other extraction-free (and saliva-based) PCR assays are available with a seemingly higher sensitivity I wonder what the added value of the current assay is for other labs. I urge the authors to 1) determine the limit of detection of their assay (see example guidelines for LOD here: https://www.fda.gov/media/135659/download) and 2) to add a “limitations of the study” section to their discussion to provide a rational for other labs to use this assay (as compared to other more sensitive extraction-free and non-swab based PCR assays).

Reviewer #3: What is the purpose of Figure 4? It's unclear, lacks a time line, and is adequately handled in the text. Consider omitting.

7. PLOS authors have the option to publish the peer review history of their article (what does this mean?). If published, this will include your full peer review and any attached files.

Reviewer #1: **Yes: **Peter M. Keller

Reviewer #2: No

Reviewer #3: **Yes: **Sheldon Campbell

---

## [Author Response · Author response to Decision Letter 1]

14 Dec 2020

Reviewer #2: I would like to thank the authors for revising their manuscript. The authors have partly addressed my comments, and I still have 2 remaining comments:

1. In their responses to the other reviewers and my comments, the authors state that their proposed protocol is for SARS-CoV-2 surveillance. However, I think the authors have mixed up the meaning of surveillance and screening, which refer to different types of tests in epidemiology (https://www.fda.gov/medical-devices/coronavirus-covid-19-and-medical-devices/covid-19-test-uses-faqs-testing-sars-cov-2). Surveillance is applied to the population level and not to report results back to the individual level, while screening refers to testing of individuals without symptoms. Thus, the authors should use the proper terminology throughout their manuscript.

We thank the reviewer for clarifying for us. We now talk exclusively about screening (and not surveillance) in the manuscript.

2. In my previous review I commented on the lack of an extensive sensitivity analysis for the current protocol. I agree with the authors that screening assays may need a different level of sensitivity as compared to diagnostic assays. However, sensitivity analysis is important to enable comparisons between assays to help labs make a decision for an assay. For rapid tests and at-home test I think lower sensitivity is justified, but considering that the current assay is PCR-based and that other extraction-free (and saliva-based) PCR assays are available with a seemingly higher sensitivity I wonder what the added value of the current assay is for other labs. I urge the authors to 1) determine the limit of detection of their assay (see example guidelines for LOD here: https://www.fda.gov/media/135659/download) and 2) to add a “limitations of the study” section to their discussion to provide a rational for other labs to use this assay (as compared to other more sensitive extraction-free and non-swab based PCR assays).

1) We have now performed the experiment suggested by the reviewer and we write the following in Results: “Next we tested the smallest number of SARS-CoV-2 RNA molecules that can be detected in the assay. We find that six RNA molecules can be detected in 19 out of 20 replicate reactions, which corresponds to limit of detection of 3,000 RNA molecules per milliliter of mouthwash (Suppl. Table 2).”

Since we were able to detect only six RNA molecules the assay is very close to the theoretical limit of detection of single viral RNA molecules. The factor that limits the experimental limit of detection is the volume of the mouthwash that we can use in the assay without inhibiting the reaction. This limits also other extraction-free assays. We are not aware of any other direct assays that are substantially more sensitive.

2) We have added a last section called “Limitations and Practical Considerations” after the Discussion section (line 196 ff). It discusses the lower sensitivity and the usefulness of the assay and draws attention to these issues more than when this discussion was embedded in the Discussion. 

Reviewer #3: What is the purpose of Figure 4? It's unclear, lacks a time line, and is adequately handled in the text. Consider omitting.

We agree that the figure is simplistic. Yet, we find it useful for illustrating a practical scheme as it implemented currently in our institute and at the retirement home. If possible, we would like to keep in the manuscript.

---

## [Editor Report · Decision Letter 2]

17 Dec 2020

A direct RT-qPCR approach to test large numbers of individuals for SARS-CoV-2

PONE-D-20-23519R2

Dear Dr. Maricic,

We’re pleased to inform you that your manuscript has been judged scientifically suitable for publication and will be formally accepted for publication once it meets all outstanding technical requirements.

Kind regards,

Ronald Dijkman, PhD

Academic Editor

PLOS ONE
---

## [Editor Report · Acceptance letter]

21 Dec 2020

PONE-D-20-23519R2 

A direct RT-qPCR approach to test large numbers of individuals for SARS-CoV-2 

Dear Dr. Maricic:

I'm pleased to inform you that your manuscript has been deemed suitable for publication in PLOS ONE. Congratulations! Your manuscript is now with our production department. 

Kind regards, 

on behalf of

Dr. Ronald Dijkman 

Academic Editor

PLOS ONE